# Remediating and Reusing Abandoned Mining Sites in U.S. Metropolitan Areas: Raising Visibility and Value

**Dona Schneider * and Michael R. Greenberg**

Bloustein School of Planning and Public Policy, Rutgers, The State University of New Jersey,
New Brunswick, NJ 08901, USA
*   Correspondence: donas@rutgers.edu; Tel.: +1-609-273-4698

**Abstract:** Abandoned mining-related sites present threats to human health and the environment, while also being potentially valuable places for redevelopment. This paper examines whether successful sustainable redevelopment is more likely in metropolitan areas, and identifies site and population characteristics that make redevelopment more likely. We abstracted data on 143 abandoned mine sites from the U.S. EPA's Superfund list, including information on site history and characteristics, remediation efforts and any continued contamination risk. Forty-one sites were located in metropolitan areas, and these underwent further document review. The EPA's updated 2002 EJScreen database was used to identify populations at risk. Data were analyzed using matched pairs and discriminant analysis statistical tests. Follow-up studies of selected sites confirmed cleanup status and plans for sustainable re-use. We found that sites located in metropolitan areas were more likely than those in non-metropolitan ones to have undergone remediation and redevelopment. Multi-use sites were more likely to have completed remediation compared to single-use sites. A combination of site and population characteristics predicted the extent and type of redevelopment at most sites. It is likely that public pressure related to human and environmental health risks and high land values serve as an impetus for the remediation and re-use of abandoned mine sites in metropolitan areas.

**Keywords:** abandoned mines; contamination; redevelopment; remediation; sustainable re-use





## 1. Introduction

The purpose of this paper is to examine the remediation and redevelopment of a set of abandoned mining-related sites in the United States. Some of these sites are in metropolitan areas, are hard to ignore because they potentially threaten human and ecosystem health, and provide potentially valuable land for redevelopment. In other words, they are visible and apparent to elected officials, businesses and community members. In contrast, similar sites in non-metropolitan areas are more likely to be both physically and politically invisible to the public, making it less likely they will be remediated and re-used [1–4]. The working expectation of this study is that abandoned mining sites in metropolitan areas are more likely to be remediated and reused than similar sites in non-metropolitan areas.

The literature tells the story of both the United States' and international efforts towards the challenges in redeveloping abandoned mining sites. Several articles provide an excellent base for our study [5–9]. For example, Kivinen characterizes mining in Finland as a "temporary" land use [5]. Mines provide essential raw materials, yet they epitomize high economic risk when the resource becomes less pure, cheaper sources of the resource are found, new processes eliminate the need for the resource, and communities pressure elected officials and companies to eliminate risks to human and environmental health.

Many mines have short economic life spans, yet their post-closure impact can endure for generations. The closed facilities may continue to pollute local water supplies, release airborne toxins, and leave slag piles, pits and dumps that contaminate the soil and deface the landscape. Ghost towns are a common legacy. One example is that of Climax Colorado.

Located at over 11,000 feet (3353 m) above sea-level in the Rocky Mountains, three quarters of the world's molybdenum supply once came from Climax [10]. Molybdenum increases the strength, hardness, electrical conductivity and resistance to corrosion of steel and other products. The town originally had a small population (estimated at 1500), including its own post office and hospital built on top of the molybdenum mine. Climax was the only U.S. community with an annual average temperature below freezing. When the settlement became a detriment to mining, the population was relocated and mining stopped for 17 years, resulting in Climax becoming a ghost town. When the mine reopened in 2012, the second author visited, noting the company had built an intricate water treatment system to capture drainage from the site and prevent it from polluting water supplies. Despite the population being gone, the legacy from the mine continues. The company is trying to get the water quality standard for molybdenum weakened, but Colorado health officials oppose altering the standard [11]. Climax, in short, exemplifies the benefits and long-term legacies of mining activities.

Reclaiming mine lands should increase public safety, reduce environmental impacts, and open up opportunities for redevelopment. Thus, considerable attention is now focused on remediating and reusing abandoned mine sites. Tyson [12] states that there are well over one million abandoned mines worldwide, including more than 500,000 in the United States. After warning readers about the dangers of entering mines, he asserts that "with a bit of planning and ingenuity, mines can be put to some pretty amazing uses." Examples of mine reuse are for energy generation; growing mushrooms, medicinal plants and some foods all year round; conducting biology, chemistry, geology, and engineering experiments; and as underground data storage facilities. Australia, Canada, China, Romania, South Africa, the United Kingdom and the United States have also cleaned up abandoned mine sites to be used as locations for hotels, golf course, lakes, and sporting events.

Researchers have suggested other creative approaches for reusing abandoned mines. For instance, Choi and Song promoted the use of abandoned mines in South Korea for photovoltaic cells. Open-pit mines are among the most difficult sites to reuse [6], yet Apostu, Lazar, and Faur developed models and procedures to evaluate open-pit mines so that some could be used as amphitheaters, off-road vehicle circuits, lakes, and landfills [7]. Mehta et al. considered the idea of reusing products recovered from abandoned mines in Gorno, Italy [8]. The context of this last paper is the idea of a circular economy, which implies sharing, leasing, reusing, repairing, refurbishing and recycling existing materials and products as much as possible [9]. The literature offers many ingenious approaches to reusing mines [13,14].

The U.S. Nuclear Regulatory Commission characterized the alternative energy path as "a breath of fresh air" for America's mine lands, and the Biden Administration has created a USD 500 million program to transform mines into places where clean energy can be sited [15,16]. The transformation will take considerable time, partly because these areas have been neglected and abandoned for so many years. For example, Pennsylvania is slated to receive USD 266 million for mine land transformation [17]. Fredericktown (population about 400) in southwest Pennsylvania is specifically mentioned as part of this effort. Located along the Monongahela River, the area is beautiful and a good place for hiking and canoeing. However, rehabilitation of the local economy will take many years, as the median age in Fredericktown is 42 years and the population is declining. Federal grants are welcome but will not be sufficient to return the area to economic health. In contrast to the efforts to rehabilitate abandoned mine sites, the U.S. Department of Labor is concerned about keeping people out of the areas. The agency launched a "Stay Out, Stay Alive" campaign, warning potential intruders of cave-ins, subsidence, fires, hazardous materials such as blasting caps and industrial explosives, hidden mine shafts, low ceilings, and other hazards [18].

In the United States, the Environmental Protection Agency (EPA) is charged with protecting people and the environment from significant environmental health risks. The agency has overlapping responsibilities: (1) it manages the National Priority List (NPL)

of Superfund sites; (2) it plays a pivotal role in safely remediating and reusing sites, and (3) it has a role in promoting clean energy [19–21]. As of July 2022, the Superfund program reported 1877 sites in progress, deleted or proposed. A few sites were listed as Superfund Alternative Agreement (SAA) sites, which means agreement has been reached with the responsible parties, thus saving everyone the cost of protracted legal cases. Part of the EPA's challenge is to hold the parties responsible to pay for remediating contaminated sites. Those efforts, however, make the Superfund program a target for attack by business interests, who feel disproportionately economically penalized [22].

Of the 1877 sites listed on the Superfund list, 143 are classified as abandoned mine lands (AMLs), defined by the EPA as "those lands, waters, and surrounding watersheds where extraction, beneficiation or processing ores and minerals has occurred." These 143 are a specific set of high-risk mining sites. We know quite a bit about these sites because Superfund mandates establishing a file for each one, including its history, contaminants of concern, cleanup decisions, stage of remediation, current status and redevelopment, if any. The database is searchable by site, and despite the fact that not all the information is entirely up to date, the information available is far greater than is available for most U.S. mining sites. For example, in 1981, the EPA named the first 114 Superfund sites [23]. At the top of the list was Commencement Bay, Washington. This site is closely identified with the American Smelting and Refining Company (ASARCO), particularly for its massive smelter and the city of Tacoma. It was placed on the NPL in 1983, with seven areas requiring remediation, including emergency actions. The EPA's files for this site are detailed, including the fact that hundreds of businesses now use the remediated area, including the massive adjacent port. The EPA has been transparent about advocating for remediating and then reusing many hazardous Superfund sites. The agency's data show that 135 Superfund sites were being reused in 2011, and that number increased to 650 in 2021. The number of jobs associated with these redevelopments increased from about 24,000 to 246,000 over two decades [24].

The actual number of abandoned mines in the United States is difficult to determine. A U.S. General Accountability Office (GAO) report gives an estimate of at least 140,000 abandoned hardrock mine features, such as tunnels, on lands under their jurisdictions [25]. Of these, almost half pose physical safety hazards, including injury or death. The GAO also reports there may be about 390,000 sites not captured in the various U.S. government databases. Little is known about many of the sites. For instance, the State of New Jersey's Geological and Water Survey Group reported approximately 450 underground mines in the state, noting that in many cases, very little information was compiled about these mines [26]. The agency created a map archive of these sites so residents can find them. In 1997, the EPA published a hardrock mining framework intended to allow its regional offices to effectively use the agency's limited national resources, including 14 recommendations and 10 action times [27]. The document served as a call to action regarding long-neglected abandoned mine concerns, to get EPA headquarters and the ten regions to gather data and create a plan to move the framework forward. Notably, the report does not mention an estimated number of sites.

The SRA conducted a case study of 20 AML Superfund sites for the EPA, finding that the large number of sites and the massive size of some made it difficult to develop communications protocols and consensus plans for their redevelopment [28]. They recommended the use of local information centers, the establishment of toll-free phone lines, the broader distribution of site-related information, the finding of alternatives to public meetings, establishing rapport with local media, and taking other steps to increase community involvement. However, 19 of the 20 sites were in rural areas.

Overall, the literature shows a dearth of data on abandoned mines, making it hard to plan a course of action, especially in isolated areas. Communities may not be aware of the threats, nor be attuned to plausible redevelopment opportunities. Turning this sentence on its head implies that metropolitan communities and their elected officials and businesses may be more likely to be aware of the hazards posed by abandoned mines and the opportunities remediation and reuse might bring to their economies. In other words,

we expect that abandoned mines would be more likely to be remediated and redeveloped in metropolitan areas than in rural ones.

In order to test this expectation in the U.S. context, we needed a dataset of abandoned mine sites identified as dangerous across the country, with sufficient information to answer the following three categories of questions:

1.  Metropolitan locations—Are EPA's AMLs in metropolitan areas more likely to have been redeveloped than those AMLs not located in metropolitan areas? The expectation is that because of a larger population at risk and potentially more valuable land, visible metropolitan area sites would be more likely to have been remediated;

2.  Site conditions and redevelopment—What proportion of metropolitan AMLs have been redeveloped into new land uses (industrial, residential, commercial, park and other)? What are the site characteristics of those AMLs that have undergone multi-use redevelopment compared to those that have not? Our expectation was that urban sites in heavily populated regions would be more likely to have a range of land re-use types and would have taken steps to make them safe;

3.  Demographic, environmental and community characteristics—What are the demographic and environmental attributes of the communities immediately surrounding the former mining sites? How do these compare to their host states and counties? What community characteristics are most associated with the current on-site land uses? The expectation was that areas within a mile of a metropolitan AML Superfund site would manifest more evidence of social and environmental justice challenges than their host counties and states.

## 2. Materials and Methods

This paper focuses on the 143 sites included in the EPA's AML Superfund program. These are highly politically visible sites because of their NPL status or SAA listing. Of the 143 sites, 47 were located in metropolitan statistical areas. Close review of the data showed that 6 of the 47 sites had no resident population within 1.5 miles. Accordingly, they were reallocated to the non-metropolitan site group. Of the 41 remaining metropolitan sites, 18 (44%) are in the west, 12 (29%) are in the northeast, 6 (15%) are in the south, and 5 (12%) are in the midwest. Almost half are in four states: Utah (n = 6), New Jersey (n = 6), and four each in Colorado and Pennsylania.

The EPA Superfund files provide rich commentaries on the listed sites, but the information is not consistent and most of the variables listed on the files are dichotomous. For example, site-specific web pages provide the site's name, EPA region, its EPA identification number, its NPL status (proposed, final or deleted), and whether the site is SAA (yes/no). There is a site homepage with a list of contacts; a description about cleanup activities and progress; a section on health and environment; a list of redevelopment activities, and site documents. Within the health and environment section, there is a list of contaminants of concern (COC). Here, the EPA identifies people and ecological resources that could be exposed to the contamination found at the site; determines the amount and type of contaminants present; and determines the human health or ecological effects that could result from contact with the contaminants. Each COC is listed, along with whether it is in groundwater, surface water, soil or solid waste. The web page also identifies whether human exposure to the COCs is under control (yes/no), whether groundwater mitigation is under control (yes/no), whether construction for the cleanup of the entire site is complete (yes/no), and whether the site is ready for reuse (yes/no). While a rare element might be mentioned as a contaminant, if it was not listed as a COC, it was not abstracted for this analysis. The most commonly listed COCs listed appear in Table 1, along with the full list of 22 site characteristics abstracted from the EPA Superfund files.

**Table 1.** Site, demographic and environmental characteristics of U.S. abandoned mine land sites on the Superfund list.

| Metric | Data Source |
|---|---|
| Site Characteristics (n = 22) | |
| Former land use: 1 = yes, 0 = no<br>● Mine<br>● Smelter<br>● Processing<br>● On-site storage | |
| Superfund site listing 1989 or earlier (earlier implies more serious threat): 1 = yes, 0 = no | |
| Primary contaminants of concern: 1 = yes, 0 = no<br>● Arsenic<br>● Cadmium<br>● Copper<br>● Lead<br>● Mercury<br>● Nickel<br>● Radium<br>● Zinc | U.S. EPA Superfund Files |
| Population relocated during remediation: 1 = yes, 0 = no | |
| Remediation projects completed: 1 = yes, 0 = no | |
| Presence of contamination in: 1 = yes, 0 = no<br>● Air<br>● Land<br>● Water environments | |
| Redeveloped land uses on site: 1 = yes, 0 = no<br>● Industrial and related commercial<br>● Light commercial or government<br>● Residential<br>● Other, including parks and roads | |
| Demographic Characteristics (n = 7) | |
| ● People of color, %<br>● Low income population, %<br>● Population with less than a high school education, %<br>● Limited English speaking family, %<br>● Unemployment rate, %<br>● Population less than 5 years old, %<br>● Population greater than 64 years old, % | U.S. Census, American Community Survey, 2020 |

**Table 1.** *Cont.*

| Metric | Data Source |
|---|---|
| Environmental characteristics and LULUs, 2016–2022 estimated by EJScreen (n = 10) | |
| Medically underserved (%) | U.S. Health Resources and Services Administration (survey data measuring lack of primary care providers, high infant mortality, high poverty rate, and high elderly population) |
| Population density (pop/mi$^2$) | U.S. Census, American Community Surveys, 2016–2020 |
| Particulate matter (PM 2.5 in µg/m$^3$) | Derived from a combination of EPA air monitoring and modeling data |
| Diesel PM (in µg/m$^3$) | Estimated from the 2017 Air Toxics update |
| Air toxics respiratory health indicator (percentiles) | Estimated for multiple air toxics by summing chronic noncancer hazard quotients (HQs) for individual air toxics. |
| Traffic proximity and volume (daily traffic count/distance to road) | Annual average vehicle counts at major roads within 500 m of block centroid divided by distance in meters, 2019. |
| Risk Management Program (RMP) facility proximity (facility count/km distance) | Sites within 5 km or nearest one beyond 5 km divided by distance in km, 2019. |
| Hazardous waste proximity (facility count/km distance) | Count of Treatment, Storage, and Disposal (TSDF) facilities within 5 km or nearest one beyond 5 km divided by distance in km, 2019. |
| Underground storage tanks, counts | Number of leaking underground storage tanks weighted by EPA, 2019, within 1500 feet of the centroid. |
| Housing built before 1960, % | U.S. Census files |
| Local Community Characteristics from Niche, Inc. (n = 8) | |
| Overall rating (scale 3–11) | C− = 3, . . . A+ = 11 |
| Jobs (scale 1–10) | D−/D = 1, . . . A = 10 |
| Schools (scale 1–11) | D−/D = 1, . . . A+ = 11 |
| Safety and crime (scale 3–8) | C− = 3, . . . B+ = 8 |
| Good for Family (scale 3–11) | C− = 3, . . . A+ = 11 |
| Outdoor quality (scale 4–11) | C = 4, . . . A+ = 11 |
| Ethnic and income diversity (scale 1–11) | D−/D = 1, . . . A+ = 11 |
| Housing (scale 1–9) | D−/D = 1, . . . A− = 9 |

Table 1 also shows the seven demographic and 10 environmental characteristics for the 1-mile and 3-mile radii around the sites, their host counties and host states, variables obtained through the U.S. EPA's EJScreen [29]. The seven demographic measures were chosen to identify places with social justice issues (proportion of people of color, poor, without a high school diploma, with limited English proficiency, unemployed, and younger and older ages). These also serve as markers for relatively vulnerable populations.

Three environmental indicators from EJScreen (particulate matter 2.5, diesel particles and air respiratory values) measure the ambient air quality surrounding each site. Note that these indicators are only as representative as the density of monitoring stations in each area and the models estimating the values between them. In many locations, air monitors are located miles away from the study sites, and located between hills and valleys. Therefore, the air quality numbers must not be taken at face value. The traffic proximity,

hazardous waste, leaking underground storage tank and risk management plan metrics measure potential contamination. For these variables, the EPA's GIS software counts the number of sites within a specified distance of the centroid of the study site, and estimated relative exposure.

The final three environmental indicators obtained through EJScreen are popuation density, medically underserved populations and housing age. Density is a surrogate for potetial demand for site re-use as well as being a surrogate for risk perception of nearby residents. In other words, high density should be associated with a greater likelihood of remediation and redevelopment.

The U.S. Health Resources and Services Administration uses survey data to estimate medically underserved populations (absence of primary care providers, high infant mortality, high poverty rate, and high elderly population). This serves as another social justice indicator. Housing age is in the EPA database as an indicator of potential lead exposure. An area with a great deal of housing built before 1960s is likely to have more houses with lead paint. However, it is also possible to use this metric to measure housing investment. The hypothesis is that relatively little new housing has been built in areas with AMLs.

The final indicators on communities come from Niche, Inc., a company that rates places and schools in the United States. Niche relies on U.S. census data, FBI and other federal government data, and also conducts its own surveys. It rates counties, cities, and selected neighborhoods. The number of places Niche rates is impressive, which makes it the only source that covers tens of thousands of locales. The six characteristics listed from Niche have scores similar to school grades, that is, ranging from F to A+. We converted these into numeric scores, as indicated in Table 1.

Niche reports describe their data sources and their statistical methods [30,31]. The second author has used Niche data in previous studies and tested its results against census data. These studies show statistical links between Niche scores, and environmental and social justice metrics. For example, Greenberg [32] studied 16 of the first Superfund sites in the United States, finding that the most stigmatized sites had significantly poorer rates for protecting against crime, family attractiveness, and Niche's overall quality rating.

A final comment on our AML dataset is that the EPA files provided sufficient public information to allow this study to occur. They also gave us enough information to classify each site into four types of redevelopments: new industrial (not a continuation of the previous industry), light commerce (stores, warehouses, storage facilities), residential (not a continuation of previous housing units) and other, such as parks, trails, wildlife habitats and other recreational activities (1 = yes, 0 = no). Our original intent was to use quantitative data from EPA's site reports for these and other considerations. We did do that, but without coming away with the precise data we had hoped to find. Some sites have extremely detailed reports, including an incredible treasure chest of data, old site maps, and notes from field operatives. Others do not. To be consistent across the sites, we had to default to the less detailed databases. This is not meant as a criticism of EPA and its contractors. Rather, we believe it to be a reality of the Superfund program, which varied in is intensity among the ten U.S. EPA regions. Each regional administrator had considerable power to make decisions.

The indicators we extracted from the EPA reports are almost all simple dichotomies (had a smelter or not, was concerned about radium or not, and so on). Some reports, for example, had information about the underground flows of contaminants, whereas others did not. In 2023, climate may now be critical, but it was not considered so when these decisions were made. Logic suggests that local climate was considered. The evidence, however, is not there. We assumed it is blended in the record of decision. Regarding the contaminants, eight specific ones were mentioned often enough in the reports to be used as dichotomous indicators (Table 1). Cyanide was not one of them, but in preparing the case reports, if cyanide was mentioned as an important concern, we included it in our analysis.

A final issue was the inability to test the expectation that more public pressure was brought to bear on the urban sites compared with the rural ones. The EPA's notes varied

in depth, rarely addressing media coverage or public protests, although we were able to find several by searching. The literature supports the idea of greater pressure in urban areas [33,34]

Before presenting the results, we note that we faced several challenging statistical issues. The most demanding was that we had 41 cases and 45 variables. The more variables per case, the more likely it was that random chance would lead to significant associations. To respond to this challenge, we used several methods to examine our expectations. For example, we compared demographic and environmental indicators in areas immediately surrounding the 41 sites with their host states, host counties, and 3-mile radius areas. To minimize false positives, we used both parametric and non-parametric (Wilcoxon signed ranks) statistical tests to compare matched pairs. As they yielded similar results, only the parametric results are presented in the text.

Questions 2 and 3 required using variables that were dichotomous, ordinal and interval. With only 41 cases, we created three redevelopment groups (0 = no redevelopment, 1 = industrial redevelopment, 2 = residential, light commercial and park land uses). A larger number of redevelopment options was possible, but this would challenge the statistical tools when there was only four to six cases in each group. With only three groups, each group had at least a dozen sites. We selected discriminant analysis to differentiate among our three groups. The method finds the underlying set of predictor variables that have the best statistical association with each of the three groups. The statistical associations are then turned into equations used to calculate values for each of the 41 cases, and average values are then calculated for the three groups. The values for each site are then compared to the average values of the groups. This method predicts a group for each case by measuring how far it is from the three groups' average values. For example, we know that hypothetical case A is an industrial redevelopment site based on the EPA AML dataset. The model finds that case A's values almost fall on the industrial site group average, whereas those of case A fall far from the values for the no redevelopment and muti-use development groups. In contrast, case B, which is also an industrial redevelopment site, gave values closer to the no-reuse group average. Case B was incorrectly predicted. A big advantage of this tool is that the user can add predictor variables and observe how the predictions change for the better or worse. For our first discriminant analysis, we used only site-specific information; in the second, we added community data.

While discriminant analysis predicts the accuracy of the groups presented by the users, it also tells the user about the correlation between each of the three groups and the predictor variables. With three groups, the method creates two new statistical "functions" (n − 1 groups). The user can determine the drivers of each of these functions by examining the correlation coefficients between the predictor variables and the two functions. The function names are based on the pattern of the correlations. Overall, two sets of outputs must be examined to explain the discriminant analysis results obtained by this study.

## 3. Results

### 3.1. Question 1: Metropolitan and Non-Metropolitan Comparisons

In total, 38% (54 of the 143) of the AML sites on the EPA's Superfund list have undergone redevelopment. For the non-metropolitan AML sites, only 26% (27 of 102) had some form of redevelopment. Most common for the non-metropolitan sites were new trails, recreational fields, nature preserves, golfing, hunting, snowmobiling and other seasonal recreational activities. Institutional controls (zoning, fencing and other restrictions) restricted access to many of the sites, and residential uses were typically not permitted. Some of the sites had local public meetings to agree upon future uses that would be subject to remediation that satisfied the EPA. These non-metropolitan AML sites had a median population of 1419 people within a 1-mile radius; their county median population was 30,052. Those in the metropolitan region group had many more nearby residents.

In strong contrast to the non-metropolitan AML sites, 65% (27 of the 41) of the metropolitan ones had at least one industrial, light commercial, residential, park or other

new land use. Fourteen sites had no re-use reported. In total, 7 of the 41 remediated sites had four types of new land uses, 5 had three of the four types, and 6 sites had two of the four types. Further exploration of the files found that light commerce, residential, and parks/roads were often co-located. With some important exceptions, industrial re-developments were solely located. In contrast to the non-metropolitan AML sites, the metropolitan sites had on average about 18 times as many people. In short, AMLs in metropolitan areas were far more likely to have been redeveloped than those located in non-metropolitan areas.

### 3.2. Question 2: Site Characteristics and Redevelopment

Discriminant analysis (using stepwise insertion) examined the relationship between the site characteristics of 41 metropolitan AMLs and redevelopment status (see the Appendix A for more details on the discriminant analysis). Since we created three categories of redevelopment, the discriminant analysis created two functions. The numbers in columns two and three of Table 2 are correlations. The names of the functions were determined by the authors based on the group most defined by the correlations.

**Table 2.** Results of the discriminant analysis for predicting the development of 41 metropolitan AML sites using site characteristic metrics.

| Site Characteristic | Function 1: Industrial Land Uses | Function 2: Residential, Light Commercial and Park Land Uses |
|---|---|---|
| Zinc contamination of concern (1 = yes, 0 = no) | 0.540 | |
| Former mine (1 = yes, 0 = no) | −0.473 | |
| Lead contamination of concern (1 = yes, 0 = no) | 0.470 | |
| Former smelter (1 = yes, 0 = no) | 0.445 | |
| Population relocated as a result of contamination (1 = yes, 0 = no) | 0.263 | |
| Remediation projects completed (1 = yes, 0 = no) | | 0.500 |
| Superfund site listing 1987 or earlier (1 = yes, 0 = no) | | −0.483 |
| Presence of air pollution a major concern (1 = yes, 0 = no) | | 0.257 |
| Canonical correlation r | 0.492 | 0.446 |

The first discriminant function in Table 2 identifies with the characteristics of sites with industrial redevelopment. Sixty percent of these AMLs historically had smelters, typically for zinc and lead. The second function identifies sites with redevelopment into commercial/residential/park and related land uses. The EPA reports that 73% of these multi-use sites had completed on-site remediation, and many were added to Superfund after 1998. Some were never listed as NPL sites, and instead were listed as SAA (voluntary agreement to remediate).

While these results are informative about the likelihood of redevelopment, the discriminant analysis accurately classified only 60% of the cases, which is better than random assignment, but not much better. Two-thirds of industrial sites were accurately classified, compared to 56% of the other two groups (multi-use and no re-use).

### 3.3. Question 3. Site and Community Characteristics of 41 Metropolitan AMLs

We next evaluated the demographic and environmental attributes of the communities immediately surrounding the redeveloped AMLs, and compared them to populations within a 3-mile radius, their host counties and host states. We expected to find

that proximate distance to an AML site would be related to social and environmental justice challenges.

We present the results of the analysis for question three in two stages. The first set of results in Table 3 are the parametric results obtained by comparing matched pairs between the 1-mile radius areas immediately surrounding each of the 41 metropolitan AML sites and those of their surrounding 3-mile radius, host county and state regions. In order to simplify the results, all the values in Table 3 are presented as ratios. A value of 1 means the 1-mile radius areas as a whole are the same as their host state, county and surrounding 3-mile radius areas. For example, the 1-mile radius areas had an average air toxics respiratory health indicator almost 40% higher than the host state, and the proportion of homes built before 1960 was 26% higher. Overall, the first column in Table 3 shows that residents within 1 mile of the sites consistently demonstrate higher levels for all of the indicators compared to their host states, and the values are significantly higher for 10 of the 16 indicators (<0.05).

**Table 3.** Ratios of demographic and environmental characteristics proximate to the 41 metropolitan AML sites compared to their state, county and 3-mile radii.

| Indicator | 1-Mile Radius Area [a,b] | 1-Mile Radius Area Compared to County [b] | 1-Mile Radius Compared and 3-Mile Radius Area [b] |
|---|---|---|---|
| Demographic metrics (n = 7) | | | |
| Unemployment rate, % | 1.23 ** | 0.95 | 0.95 |
| Population of color, % | 1.23 ** | 1.02 | 1.00 |
| Population with less than a high school diploma, % | 1.23 ** | 1.05 | 1.00 |
| Limited English language proficiency, % | 116.6 * | 0.82 ** | 0.95 |
| Low income population, % | 116.2 * | 1.06 | 0.98 |
| Population greater than 64 years old, % | 113.6 * | 1.11 ** | 1.08 |
| Population less than 5 years old, % | 104.2 | 0.90 | 1.35 |
| Environmental metrics (n = 9) | | | |
| Air toxics respiratory health indicator | 139.8 ** | 0.93 | 0.97 |
| Housing built before 1960 | 126.0 ** | 1.11 * | 1.02 |
| Risk Management Program facility proximity | 115.2 * | 0.84 ** | 1.06 |
| Traffic proximity and volume | 115.2 * | 0.83 ** | 0.87 ** |
| Hazardous waste proximity | 111.4 | 0.85 * | 0.94 |
| Leaking underground storage tanks | 110.4 | 0.81 ** | 0.90 * |
| Diesel PM | 109.0 | 0.92 | 0.98 |
| Particulate matter (PM 2.5) | 105.8 | 0.95 | 0.91 |
| Medically underserved | NA | 1.30 * | NA |

[a] Numbers are compared to host state. [b] Ratios are normalized to 1.0 to simplify interpretation of the results. Significance reports of two-sided matched pair tests, ** $p < 0.01$, * $p < 0.05$ (confirmed with Wilcoxon signed ranks test).

The results in the third and fourth columns do not show a simple pattern. In fact, there are more ratios below 1 than above 1, but the ratios do tend to hover around 1. The major takeaway from these two columns is that the outcomes with regard to redevelopment are influenced by factors outside the immediate 1-mile radius of the sites. For example, the 1-mile and 3-mile radius results are not significantly different for 13 of the 15 indicators.

Our first discriminant analysis using the characteristics of each site correctly classified 60% of our 41 cases. However, when we added community characteristics to the site

characteristics in our second discriminant analysis, the proportion of the 41 cases accurately classified rose to 87%. Table 4 reports the results of that analysis. For AMLs with new industrial land uses (function 1), the method identified that the newly redeveloped industrial sites are near or attached to existing hazardous waste facilities. A disproportionate share of people of color live nearby; their housing stock is older (which may imply little new home building) and their host communities' safety and crime ratings are relatively low. In other words, adding community metrics increased the method's ability to correctly identify sites with new industrial development from 66% to 93% accuracy.

**Table 4.** Results of the discriminant analysis for predicting the development of 41 metropolitan AML sites using site and surrounding community metrics.

| Metric | Function 1: Industrial Land Uses | Function 2: Residential, Light Commercial, and Park Land Uses |
|---|---|---|
| Site characteristics | | |
| Lead contamination of concern (1 = yes, 0 = no) | 0.298 | |
| Smelter at the site (1 = yes, 0 = no) | 0.256 | 0.223 |
| Zinc contamination of concern (1 = yes, 0 = no) | 0.249 | |
| EPA reports remediation construction projects completed (1 = yes, 0 = no) | | 0.273 |
| Superfund site pre-1987 (1 = yes, 0 = no) | | −0.271 |
| Surrounding area characteristics | | |
| Hazardous waste proximity (TSDF count/km distance) | 0.528 | |
| People of color, % | 0.350 | |
| Housing built before 1960 | 0.336 | |
| Population density (pop/mi$^2$) | | 0.344 |
| Safety and crime (3–8, where 3 is lowest (C−) and 8 is highest (B+)) | −0.213 | |
| Jobs (1–10, where 1 is lowest (D) and 10 is highest (A) in this set of places) | | 0.312 |
| Good for family (3–11, where 3 is lowest (C−) and 11 is highest (A+) in this set of places) | | 0.283 |
| Ethnic and income diversity (4–10, where 4 is lowest C and 10 is highest (A) in this set of places) | | 0.270 |
| Canonical correlation r | 0.764 | 0.629 |

The second function identifies AML sites that have been redeveloped into multiple uses. The strongest metric for this set of cases is population density within a 1-mile radius of the site. The average population density of the set is 3980 per square mile, which would fall between a densely populated suburban community and a city. In contrast, the average population density for the industrial group was 2753 per square mile and only 1086 per square mile for the no redevelopment set, many of which are in rural or semi-rural locations.

While population density was the strongest correlate for predicting the multi-land use function, the second strongest correlate was the Niche rating of the job market. Reiterating, Niche is a private organization that rates tens of thousands of places in the United States. Their scale ranges from an F (no F scores given in this dataset) to an A+. We converted the alpha rating into a numeric scale (see Table 1). The average for this set was 7 (equivalent to a B), compared to 6.5 (B−/B) for the industrial site group and 5.3 (C+) for the no redeveloped land use sites. In essence, the message of these two indicators is that the

more densely settled AML sites with higher job ratings are places that have more valuable land to redevelop. Noted as well are the Niche ratings of good place to raise a family and that offer more population diversity, attributes consistent with an attractive area. Perhaps surprisingly, almost half of the multi-use sites had historic smelting activities. The discriminant analysis accurately classified 91% of the multiple land use site cases, compared to 55% in the earlier site characteristics model (Table 2).

Sites with no redevelopment were the least accurately characterized by this second discriminant analysis. This accurately predicted 75% of the cases compared to over 90% of the industrial and multi-use site groups. These sites were placed on the NPL list early in the program, were least likely to have smelters, the construction for remediation here was least likely to be completed, and many had been AMLs where slag and tailings piles remain. Notably, the no redevelopment group had among the lowest 1- and 3-mile radius populations in the full set of 41 sites. Their population densities more closely resemble the 102 sites in the non-metropolitan AML set.

Statistical accuracy is valuable; however, ground truthing is an excellent way to verify findings. We decided to examine some of the correctly predicted sites for that purpose. We selected two sites from each redevelopment group (function 1, function 2 and no redevelopment), purposefully aiming for geographic diversity. We present these below.

Case 1 (industrial): The 300-acre Alcoa smelter site is located along the Columbia River three miles northwest of Vancouver, Washington (Figure 1). The immediate surrounding population is sparse, although the site is located within a metropolitan area. The smelter operated on a portion of the site from 1940 until 1985. For almost a decade near the end of its service, operators dumped spent potliner on the ground outside of the smelter, contaminating both the soil and groundwater with cyanide and fluoride. In 1990, the site was placed on the NPL and the hazardous soil was removed. While the risk to human health was reduced, it was not eliminated from groundwater. In 1996 the EPA declared the site ready for industrial re-use and removed it from the NPL, with continued monitoring, fencing and other institutional controls. The Port of Vancouver purchased 218 of the 300 acres in 2009, locating a new 30-acre marine terminal (Terminal 5) on site. The Port continues to seek new industries for the site, listing Terminal 5 as a business opportunity. To increase the attraction, it installed a new loop track around Terminal 5 that links to a rail complex that can handle bulk commodities [35]. Even though this valuable riverfront site is linked to a transportation complex and poses limited threats to human health, it remains underutilized. Currently, the EPA lists only two businesses on the site, employing three people and generating USD 476,000 in annual sales revenue.

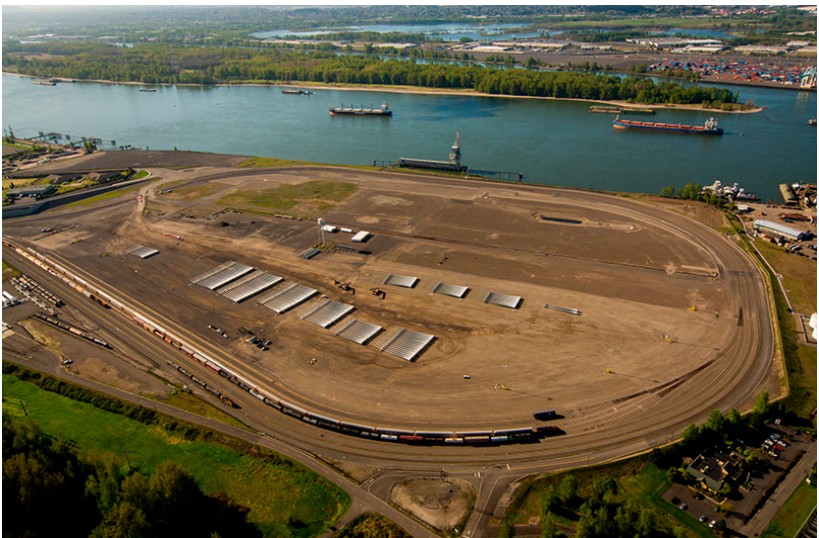

**Figure 1.** Terminal 5 with loop rail at the former Alcoa Smelter Superfund site. Source: Port of Vancouver [35].

Case 2 (industrial): The 140-acre Macalloy site is on the Charleston Peninsula in Charleston, South Carolina. Located just west of the site is a neighborhood with low-income and minority residents. From 1941 to 1998, the site functioned as a ferrochromium alloy smelter, generating tons of particulate dust and sludge that were held in an unlined surface reservoir. EPA placed the Macalloy site on the NPL in 2000. The plant installed a stormwater management system, but twice experienced outfalls to a nearby creek and wetlands area, posing concerns about multiple contaminants in soil and water. In 2002, the cleanup involved the excavation of radiological materials, on-site chemical reduction, and the enhanced chemical treatment of groundwater. In 2005, a developer created an industrial park on 30 acres in the southern portion of the site, but the remaining 110 acres remained undeveloped. In 2021, the EPA listed four industrial-use business on-site, employing 30 people and generating more than USD 6 million in annual revenue. In June 2021, the EPA announced that 134 of the 140 acres of the site were coming off the NPL. Craig Zeller, EPA Remedial Project Manager, announced that the EPA had been overly conservative in its restrictions, and that residential development would now be allowed [36]. In November 2021, Capital Development purchased the site for industrial purposes, with plans to take advantage of its location proximate to the Hugh K. Leatherman Terminal in North Charleston [37]. John Knox, CEO of Capital Development, announced a strategic partner as Shipyard Creek Logistics Center, which would create "world-class port infrastructure" [38].

Case 3 (multi-use, Figure 2): The 446-acre Midvale Slag site is located between Midvale City and Murray City, Utah. From 1871 to 1971, five lead and copper smelters operated at the site, contaminating soils and groundwater with heavy metals. The EPA placed the site on the NPL in 1991, separating the site into two operating units (OUs). OU1 was the northern 266 acres; the remaining 180 southern acres were OU2. OU2 was further separated into three parts (smelter waste and slag, groundwater and the riparian zone). The cleanup of OU1 was completed in 2006, and the EPA designated the site ready for multiple uses, including residential. The cleanup of OU2 took longer, requiring the construction of a dam and other stabilization measures. As of 2012, 20% of the Midvale Slag site was redeveloped, infusing Midvale's economy with approximately 600 new jobs and USD 1.5 million in annual property tax revenues. Redevelopment included a 95,000-square-foot grocery store, 175,000 square feet of Gold and Silver LEED-certified office space, more than 1000 completed residential units, and a light rail station with 200 parking spaces. The area, now known as Bingham Junction, serves the growing southwest region of the Salt Lake area, and includes an 18-acre park with local and regional trails [39,40]. In 2015, the EPA removed the entire site from the NPL, presenting Midvale City with an award for "Excellence in Site Reuse".

Case 4 (multi-use): The Li Tungsten site is complex, now consisting of 56 acres of waterfront in the City of Glen Cove (Nassau County), New York. The original site was 26 acres separated into the Captain's Cove property and a 4-acre wetland. The site served various industrial uses since the mid-1800s, had multiple owners and activities, but most concerningly was its use for processing tungsten and other ores into specialty metal products and dumping the wastes on site. The EPA lists contaminants of concern at the site as arsenic, lead, radium, thorium and PCBs. The agency placed the site on the NPL in 1992, launching immediate action to remove chemical storage tanks and demolish structurally unstable storage buildings. Later, contaminated soils were removed, but groundwater contamination remained. During routine dredging in 2001, the U.S. Army Corps of Engineers discovered radioactive materials in Glen Cove Creek, leading to the inclusion of the creek as part of the site. In 2016, RXR Glen Isle Partners purchased the property from the City with plans to transform the area along the creek into a 28-acre, USD 1 billion luxury residential and shopping community at Garvies Point. The plan would create 1110 residential units, a marina, parks and other amenities [42]. Other parts of the site remain restricted to light industry and commercial businesses, and some areas remain designated as Brownfields, or state- or federal-designated hazardous waste sites. Groundwater monitoring continues, and additional plans to reduce potential human exposures to residual soil contamination

include the construction of sidewalks and a municipal parking lot. In 2019, the first phase of the rental units at Garvies Point became available for leasing, and half of the 167 condominium complex units called The Beacon were pre-sold [43]. In 2023, the U.S. Army Corps of Engineers declared Glen Cove Creek a navigation asset, as it now provides a dredged channel 8 feet deep and 100 feet wide for heavy tonnage, as well as rehabilitated bulkheads supporting recreational boating and commercial marinas [44].

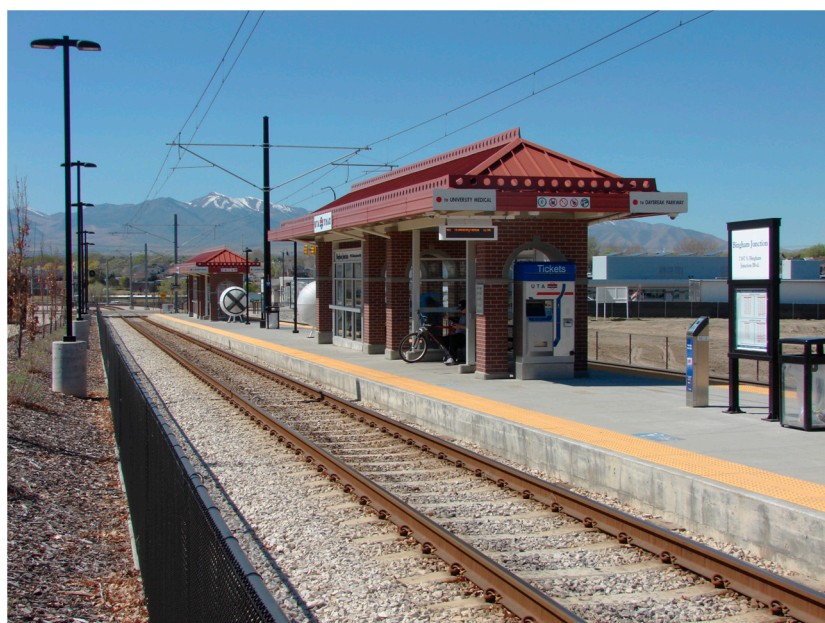

**Figure 2.** Bingham Station, serving the new residents of the revitalized Midvale Slag Site. Source: An Errant Knight [41].

Case 5 (no development): The Captain Jack Mill site (Figure 3) is a subsurface gold and silver mine located in a narrow valley called California Gulch, just south of Ward, Colorado. Mining activity took place here from 1860 to 1992, contaminating soil and surface water with metals. The EPA put the site on the NPL in 2003. One year later it removed drums of chemicals and concentrated mine wastes from the site, along with household waste, debris and paint containers. The mine tunnel was rehabilitated in 2007, allowing impounded mine water to be removed. Despite these efforts, the problem of surface and subsurface water contamination remained. A surface water plan was implemented in 2012, consisting of consolidating mine waste into two cells, covering the cells with vegetation and constructing surface water diversion structures on them. A 2013 flood event showed the system was mostly successful, with only minor impacts to the surrounding region. The subsurface remedy began in 2016, consisting of an in-tunnel water treatment system with a flow-through bulkhead, a cheaper alternative than water treatment facilities for treating acid mine drainage. In October 2018, the Colorado Department of Public Health and Environment and the EPA determined that contaminated water originating from the tunnel was responsible for a fish kill event in Left Hand Creek. Water sampling showed high acidity and heavy metals contaminating the area almost five miles downstream of the site. Emergency response and monitoring will continue while additional modifications are made to the in-tunnel treatment system. Currently, no redevelopment is planned for this portion of the site [45].

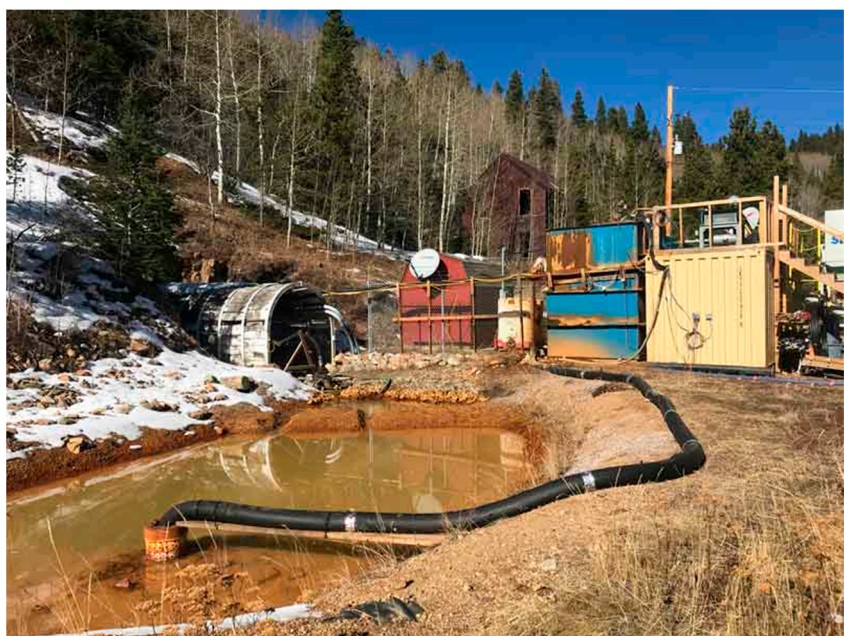

**Figure 3.** The active treatment system used at the Big Five Tunnel after the fish kill event downstream from the Captain Jack Mine Superfund site, 2018. Source: The Watershed Center [45].

Case 6 (no development): The 79-acre Foote Mineral Company site is located in East Whiteland Township, Pennsylvania. The company began processing lithium ore and other minerals for the manufacture of lithium metal, chemicals and inorganic fluxes in 1941. When the plant closed in 1991, it left behind two quarries, a solvent burn pit, a lined waste-water basin and more than 50 buildings where processing activities took place. The company had already leveled and backfilled three unlined settling lagoons when it closed. The site is located along public roads with mixed commercial and agricultural properties. It also borders a former railroad right of way and an electric utility substation. Wells within four miles of the site had supplied drinking water to more than 42,000 people, but because of contamination, these had to be taken out of service. The EPA placed the site on the NPL in 1992 and removed the underground storage tanks. The company agreed to demolish the remaining buildings in 1998 and then sold the property in 2021 to Frazer-Exton Development, L.P. After the sale, bromate contamination of the groundwater was detected, as was the low-level radioactive contamination of on-site soils. Remediation of the contamination was complete in 2017, and in 2018 the site received the EPA's designation as ready for use. The EPA mandated institutional controls, prohibiting residential uses or the installation of new wells. While no redevelopment has yet occurred, in 2023 the developer Charlie Lyddane announced a plan to build a 2 million-square-foot data center on the site at a cost of approximately USD 6 billion. While Lyddane has no specific buyer or user so far identified, the community has registered its opposition to the scope of the proposal [46].

## 4. Discussion and Conclusions

Mines spurred economic growth during the industrial revolution. Some of the world's wealthiest families have earned multi-generational benefits from mining, processing and smelting ores, and shipping minerals. Yet, the vast majority of abandoned and even operating mines, smelters, and related facilities have become eyesores stigmatized by what can be seen on the surface, and what is assumed to contaminate air, water and land near the site. Their legacy makes it difficult to attract developers. The images of danger beneath the ground, collapsed and flooded tunnels, visible slag piles, pits, and many other characteristics of tens of thousands of mines are a daunting burden to overcome.

*Sustainability* has encouraged authors to seek sustainable responses to the mining legacy. In the spirit of this journal's publications about mining, we examined 41 mining sites to demonstrate what is plausible to accomplish in metropolitan regions. Regarding the three expectations stated in the introduction, we expected and found that the EPA's mining Superfund sites in metropolitan areas were much more likely to have been redeveloped than those not located in metropolitan areas. It is not an exaggeration to assert that the 27 redeveloped sites out of the 41 metropolitan ones are poster exhibits of what is feasible. However, it is important to add that there is a dearth of information about mining lands, even among these sites that have received far more attention than others.

For the 41 metropolitan sites, we observed interesting variations among both the sites and their host states. Additional in-depth studies of the clusters of multi-use redevelopment in the Salt Lake City, Utah area, along with Colorado, New Jersey, Pennsylvania and other places might provide insights into the subtleties of what is possible for metropolitan AMLs. The U.S. government and states have been working to create conditions for redevelopment. We note that the EPA's mining lands programs have expanded, and the U.S. government has emphasized energy projects. Additionally, we found that many of the most exciting redevelopment projects include commercial, government, recreation and housing. These projects have been informed by major investments in site exploration and data development, which has become a major limitation for reusing mining lands in the United States and internationally, where the dearth of information continues to be a bottom-line problem.

We expected to find that within one mile of the 41 metropolitan AML Superfund sites, we would find more evidence of social and environmental justice challenges than in their host counties and states. Clearly, the areas around the sites showed more evidence of social and environmental injustice than their host states, but this was not the case for the host county or the 3-mile area surrounding the site.

Waste facilites, nuclear power plants and large industrial sites form a set of enduringly objectionable locally unwanted land uses (LULUs) [47]. These most noxious LULUs are essentially regarded as neighborhood cancers. Greenberg, Popper and West (1990) created the acronym TOADS (temporarily obsolete abandoned derilict sites) to describe activities that also function as neighborhood cancers, spreading their impacts across the local landscape [48]. AMLs exemplify TOADS, and efforts towards their remediation can only benefit their surrounding communities.

We applaud governments and developers trying to make AMLs and their accompanying land uses less dangerous by remediating and locating cleaner new facilties on them. It is important when doing so that they negotiate remediation efforts and new land uses with local communities. Yet, the reality is that while AMLs may be a serious problem, their lack of physical and political visibility implies that successful redevolpment will take an enormous effort on the part of elected officials and their constituencies to make this problem a priority. It is not surprising that sites with massive visible smelters, located in metropolitan areas and supported by businesses that want to avoid the Superfund process, form a disproportionate share of redeveloped sites with sustainable solutions.

**Author Contributions:** Conceptualization, D.S.; Methodology, M.R.G.; Validation, D.S. and M.R.G.; Formal analysis, M.R.G.; Investigation, D.S.; Data curation, D.S. and M.R.G.; Writing—original draft, D.S. and M.R.G.; Writing—review & editing, D.S. and M.R.G. All authors have read and agreed to the published version of the manuscript.

**Funding:** This research received no external funding.

**Institutional Review Board Statement:** Not applicable.

**Informed Consent Statement:** Not applicable.

**Data Availability Statement:** Not applicable.

**Acknowledgments:** We appreciate constructive suggestions from the reviewers and editors. The authors are solely responsible for the content of the paper.

**Conflicts of Interest:** The authors declare no conflict of interest.

**Appendix A**

Discriminant analysis is used to classify people or places into one of a group of user-defined categories. The tool allows the user to measure which variables help explain their classification scheme and which do not. Like linear regression, the tool describes both the association between the group classification and the predictors. Given the set of variables available, the tool then predicts which group each case is in. As our dataset of metropolitan AMLs includes 41 cases and 29 variables, the tables with the results would be too lengthy to include either in the text or in supplemental tables. Instead, this Appendix A provides an example of the method using a subset of 18 of our cases and six select variables. The results of this presentation are not meant to be taken at face value and are only meant to show some, but not all, the steps in the discriminant analysis process.

Example: Our 18 selected cases represent industrial reuse, multiple new land uses, and no redevelopment—6 from each group. We also selected a subset of six local community variables from Niche with their means and standard deviations. Table A1 shows the descriptive data for the selected variables. The highest mean score was for outdoor quality and the lowest for housing.

**Table A1.** Summary of data for local community measures for 18 metropolitan AML sites.

| Variable | Mean (1 = Lowest, 11 = Highest) | Standard Deviation |
|---|---|---|
| Cost of living | 5.5 | 2.0 |
| Family | 6.7 | 2.3 |
| Housing | 4.9 | 2.1 |
| Jobs | 6.2 | 2.6 |
| Outdoors | 8.9 | 1.7 |
| Schools | 5.9 | 3.0 |

To describe the data and the relationships, discriminant analysis uses matrix algebra to create n-1 new artificial variables called functions from the raw data. As we had three groups, the tool created two as yet unnamed functions. An eigenvalue is the characteristic root of a data matrix, an important property because it tells the user how much of the initial variation in the raw data is likely to be captured by the analysis. The higher the eigenvalues and the canonical correlations, the stronger the relationship between the original groups and the selected variables. The two eigenvalues (0.778 and 0.625) and canonical correlations (0.661 and 0.620) shown in Table A2 point to moderate relationships. If the eigenvalues were 1.4 and 1.2, we would expect stronger results with correlations > 0.8.

**Table A2.** Eigenvalues.

| Function | Eigenvalue | % of Variance | Canonical Correlation |
|---|---|---|---|
| 1 | 0.778 | 55.5 | 0.661 |
| 2 | 0.625 | 44.5 | 0.620 |

In order to understand the meaning of the two yet unnamed functions created by the tool, we examined the structure matrix, which consists of correlations between the two new variables created by the tool and the six original variables (Table A3). Note that the correlations produced rarely exceed 0.7 because the tool is correlating variables with different scales. For this example, Function 1 describes a group of cases that have good schools ($r = 0.407$) and family circumstances ($r = 0.262$) but poor job ratings ($r = -0.318$) and a high cost of living ($r = -0.225$). In contrast, Function 2 has places with low outdoor quality ($r = -0.730$) but good housing ($r = 0.404$) and a relatively low cost of living ($r = 0.257$). The user can now name the functions.

**Table A3.** Structure matrix (numbers in table are correlation coefficients).

| Predictor Variable | Function 1: High-Quality Public Schools and Family Setting | Function 2: Poor Outdoor Quality but High-Quality Housing |
|---|---|---|
| Cost of living | −0.225 | 0.257 |
| Family | 0.262 | −0.048 |
| Housing | 0.081 | 0.404 |
| Jobs | −0.318 | 0.167 |
| Outdoors | −0.020 | −0.730 |
| Schools | 0.407 | 0.080 |

At this point, the analysis could stop because we can see the major relationships revealed these few variables and the 18 cases. However, to dig deeper and predict, we calculated Fisher's linear discriminant coefficients to classify each of the 18 cases into the three groups. The Fischer's test finds vectors that maximize the separation of the means and minimize the projected within-group variance. Table A4 shows these equations. The reader can see that each group has stronger relationships with several variables and weaker ones with others.

**Table A4.** Fisher's function coefficients.

| Predictor Variable | Group 1 | Group 2 | Group 3 |
|---|---|---|---|
| Cost of living | 6.95 | 6.31 | 6.04 |
| Family | −6.05 | −5.63 | −4.94 |
| Housing | −4.56 | −3.46 | −3.473 |
| Jobs | 3.05 | 2.87 | 1.81 |
| Outdoors | 9.01 | 7.74 | 8.05 |
| Schools | 5.41 | 4.73 | 4.99 |
| (Constant) | −58.34 | −46.58 | −47.85 |

The numbers derived from applying these equations allow each of the original 18 cases to be compared by their probability of being in group 1, 2, or 3. Using the equation for each group shown in Table A4, a value is predicted for each case and compared to the average value (centroid) for each of the three groups. The predicted value is compared to the three centroids, and each case is classified based on distance from the nearest centroid. Table A5 gives the results for 4 of the original 18 selected cases, chosen to show both correct and incorrect predictions. The last column lists the Mahalanobis distance to the predicted centroid.

**Table A5.** Illustrative results for four predicted cases.

| Case | Actual Group | Predicted Group | Probability of Being in Predicted Group | Squared Mahalanobis Distance to the Predicted Centroid |
|---|---|---|---|---|
| 12 | 1 | 1 | 0.959 | 0.084 |
| 2 | 1 | 1 | 0.879 | 0.258 |
| 9 | 2 | 1 | 0.905 | 0.199 |
| 15 | 1 | 2 | 0.920 | 0.168 |

Note that cases 12 and 2 (correctly predicted) had high probabilities of being accurately predicted, and these two cases model group 1. The best prediction was case 12, with a probability of 0.959 and a distance of only 0.084 from the group 1 centroid. Case 2 was correctly predicted to be in group 1, with a probability of 0.879 and a distance of 0.258 from the group 1 centroid. In contrast, cases 9 and 15 were inaccurately predicted. Cases 9 and 15 had high probabilities of being incorrectly predicted because they are close to one centroid but were supposed to be closer to another. In short, the four cases include two almost perfectly predicted cases and two very imperfect predictions.

Recognizing that our example using 18 cases and six variables is too small for full scale analysis, we can report that the tool accurately predicted 15 of those cases and that each group had one inaccurate prediction. We found discriminant analysis to be a good choice for this dataset, but had our variables been primarily dichotomous, we would have selected logistic regression as it would be easier to explain the results.

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
