# Peer review of "Remediating and Reusing Abandoned Mining Sites in U.S. Metropolitan Areas: Raising Visibility and Value"

_sustainability, doi:10.3390/su15097080_

Round 1

Reviewer 1 Report (Previous Reviewer 2)

Dear authors,

I congratulate you on your work and appreciate that you have taken into account the initial comments (for the most part resolved in the new version of he article). Under these circumstances, my recommendation to the editors is to publish the article in its current form.

Best wishes!

Author Response

Thanks, we appreciate the support. 

Reviewer 2 Report (Previous Reviewer 3)

Abstract: The abstract of this manuscript was improved.

Introduction: The introduction was improved.

Materials and methods: All the tables and figures in this manuscript were revised to collect the order, the tables in this manuscript were easier to understand compared with the previous version. It better if the authors can provide the supplementary document for the assessment and the discriminant analysis.

Conclusions: The conclusions are already revised that easier to understand compare with previous version.

Author Response

Thank you for your keen eye on this paper.  It has been helpful.  Rather than submit a supplement which would be too lengthy for the 41 cases and 29 variables in our data set, we added an appendix with an example that is more easily understood.  It includes 18 of our cases and six selected variables, describes the steps of the discriminant analysis process for those less statistically inclined, provides tables of the results and a description of how we interpreted the findings. We hope the reviewer finds this helpful.

This manuscript is a resubmission of an earlier submission. The following is a list of the peer review reports and author responses from that submission.

Round 1

Reviewer 2 Report

Dear authors,

Your article is an interesting and up-to-date one, which brings to the fore the problem of abandoned mining sites (more precisely, it analyzes how they are reclaimed or not and the implications on the demography of the region, environmental and economic aspects).

The working methodology is very well described and implemented, and the results are clearly presented. Moreover, to support these results, 6 case studies are briefly presented, from both categories (success stories, or on the contrary, former mining areas that are still waiting for a new reuse).

The results of he study confirms the hypothesis from which you started, namely that those mining sites that are located in metropolitan areas (with greater public and political visibility) are the ones for which efforts are made for the purpose of reuse, while sites located in less visible areas remain abandoned or the degree of reuse (with the related social, environmental and economic consequences) is very low, far below their potential.

As comments, I advise you to review the article very carefully and eliminate some small errors, such as:

Line 73: Apostou must be replaced with Apostu

Line 181: revew must be replaced with review

Etc.

Also, in the Abstract you mention Conclusion, so I would suggest you to rename the last chapter: 4. Discussions and conclusions.

Finally, I congratulate you on your paper and wish you much success in the future.

Reviewer 3 Report

This manuscripts discussed about remediating and reusing abandoned mining sites in US. The topic is intersting in the field point of mining engineering. But for the readers in other fields, it would be difficult for understanding. This manuscript is requried to re-written with wel-organized and concideration for the common readers. To improve the manuscript, please find my comments and suggestions below.

- Abstract: The abstract was written separately into 1) Background, 2) Methods, 3) Results, and 4) Conclusion. Please consider to write them together with good story-telling. It would be better and attacted for more readers.

- Introduction: The contents in the introduction were also written separatelty without the connection. The authors should revise an introduction as well as summarize the literature reviews and rewrite the content which including background, literature review and relevant, research gap, objectives and indicate the expected outcomes with good story-telling, to make the easy understanding for the readers espectially in other fields, as well as gain more attraction.

- Materials and method: The materials and methods were not written well the detail of process still uncleared and the table that show the metric and data source is hard to read and understand. The authors must provide the background of correlations and make a support document how this correlations from (I recommend the authors must prepare these information in supplementary for the readers).

- Results: In the result part, this part has an answer of the question that was asked in the introduction part. The question and answer should be together for easily understanding for reader.

- The conclusion was not shown in the main text, please add the conclusion.

- Line 181: wrong typing word "revew" it should be "review", please kindly check another words in manuscripts.

- Line 200: “The numbers for air quality measures must be not be taken ” please kindly check and revise this sentence. 

Round 2

Reviewer 1 Report

Authors make a good review, carried out corrections and answered appropriately the questions. The manuscripit was sustancially improved. The manuscript is suitable for publication

Reviewer 3 Report

The authors did not revised the manuscript as per reviewer's comments and suggestion. The quality of this manuscript is not enough to be published it this journal.

1.The abstract is still written with separate parts without a connection between each of them which is not attractive to the reader.

2.The introduction in lines 80-104 is still unclear. The objective of the literature is not provided.

3.The authors did not provide the principle information about the assessment of correlations or any equation that made the reader hard to understand in assessment values that are provided in Table 2.